# Current Knowledge about Mechanisms of Drug Resistance against ALK Inhibitors in Non-Small Cell Lung Cancer

**DOI:** 10.3390/cancers13040699

**Published:** 2021-02-09

**Authors:** Elisabeth Smolle, Valentin Taucher, Joerg Lindenmann, Philipp J. Jost, Martin Pichler

**Affiliations:** 1Division of Pulmonology, Department of Internal Medicine, Medical University of Graz, 8036 Graz, Austria; 2Division of Cardiology, Department of Internal Medicine, Hospital Barmherzige Schwestern Ried, 4910 Ried, Austria; valentin.taucher@medunigraz.at; 3Department of Thoracic Surgery, University Clinic of Surgery, Medical University of Graz, 8036 Graz, Austria; jo.lindenmann@medunigraz.at; 4Division of Oncology, Department of Internal Medicine, Medical University of Graz, 8036 Graz, Austria; philipp.jost@medunigraz.at (P.J.J.); martin.pichler@medunigraz.at (M.P.); 5Medical Department III of Hematology and Oncology, Klinikum rechts der Isar, Technical University Munich, 81675 Munich, Germany; 6Central Institute for Translational Cancer Research, Technical University Munich, 81675 Munich, Germany; 7German Consortium for Translational Cancer Research (DKTK), 81675 Munich, Germany; 8Department of Experimental Therapeutics, The UT MD Anderson Cancer Center, Houston, TX 77030, USA

**Keywords:** ALK tyrosine kinase inhibitors, non-small cell lung cancer, acquired resistance mechanisms, epithelial-mesenchymal transition

## Abstract

**Simple Summary:**

Lung cancer is a devastating disease, with non-small cell lung cancer (NSCLC) being the most common subtype. With the development of novel targeted therapeutics, survival times have continuously improved over the past two decades. In a subset of NSCLC, gene rearrangements of the anaplastic lymphoma kinase (ALK), or gene fusions involving ALK, can be determined. ALK-inhibitors are increasingly used as a standard of care in patients with ALK gene abnormalities, and can also be administered as first-line treatment in advanced-stage NSCLC. However, over the disease course, cancers tend to develop resistance mechanisms, warranting the switch from first- to second- or third-generation ALK inhibitors. With this literature review, we aim to give a concise overview about these resistance mechanisms, and what kind of sequential treatment may be feasible if therapy failure upon an ALK inhibitor occurs.

**Abstract:**

Non-small cell lung cancer (NSCLC) accounts for the majority of lung cancer subtypes. Two to seven percent of NSCLC patients harbor gene rearrangements of the anaplastic lymphoma kinase (ALK) gene or, alternatively, harbor chromosomal fusions of ALK with echinoderm microtubule-associated protein-like 4 (EML4). The availability of tyrosine kinase inhibitors targeting ALK (ALK-TKIs) has significantly improved the progression-free and overall survival of NSCLC patients carrying the respective genetic aberrations. Yet, increasing evidence shows that primary or secondary resistance to ALK-inhibitors during the course of treatment represents a relevant clinical problem. This necessitates a switch to second- or third-generation ALK-TKIs and a close observation of NSCLC patients on ALK-TKIs during the course of treatment by repetitive molecular testing. With this review of the literature, we aim at providing an overview of current knowledge about resistance mechanisms to ALK-TKIs in NSCLC.

## 1. Introduction

Lung cancer is the leading cause of cancer-related deaths worldwide [1]. Approximately 80–85% of lung cancers count among non-small cell histology (NSCLC) [1], and approximately 2–7% of NSCLC cases feature positivity for anaplastic lymphoma kinase (ALK) gene rearrangement or connection with echinoderm microtubule-associated protein-like 4 (EML4) [2,3]. Fusion with the EML4 gene remains the most common form of ALK alteration [4]. NSCLC patients featuring ALK-EML4 gene fusion are very sensitive to treatment with ALK tyrosine kinase inhibitors (ALK-TKIs). Fusion of the ALK-EML4 genes in NSCLC can be detected in tumor samples by means of various methods, first of all fluorescence in situ hybridization (FISH), or else quantitative reverse transcriptase polymerase chain reaction (qRT-PCT) and immunohistochemistry (IHC) [5]. The disadvantage of RT-PCR is that it highly depends on RNA quality, which is often less than ideal in formalin-fixed and paraffin-embedded tissue samples [6,7]. However, next-generation sequencing (NGS) panels, such as the Archer^®^FusionPlex^®^ panel provide an effective alternative method for the detection of both known and novel ALK gene rearrangements with great accuracy [8]. In most specialized centers, ALK gene rearrangement analysis has become standard in the diagnostic workup of NSCLC, and ALK-inhibitors are increasingly used for NSCLC treatment in this particular subtype. According to a meta-analysis by Li et al., ALK-inhibitors were found to significantly improve the overall survival (OS) and progression free survival (PFS) in NSCLC patients, especially in patients whose tumors harbor ALK- or ROS1 gene fusions [5]. Median OS for ALK-positive NSCLC patients has nowadays increased to seven years [9,10], being currently the best reported OS of all forms of metastatic NSCLC defined by genomic variants. ALK inhibitors contributed to a better prognosis of patients, having improved one-year or two-year OS, PFS and objective response rate (ORR). Still, it has to be pointed out that ALK-positive NSCLC is a considerably aggressive subtype, mainly because of its inevitable tendency to cause brain involvement. Figure 1 shows how the EML4-ALK fusion gene is constructed (Figure 1).

That is to say, existing ALK inhibitors bind with varying contact sites in the ATP binding pocket, which results in unique and specific ways of inhibition of different ALK mutations. This was confirmed in a study on ALK mutations in neuroblastoma patients, in ALK-TKI resistant NSCLC patients and subjects suffering from myofibroblastic tumors [12]. Furthermore, the inhibition profile of different ALK-TKIs is affected by different fusion variants of EML4-ALK, or by the properties of other fusion partners [13]. In a study from Lin and colleagues from 2018, ALK variants were identified in a cohort of 129 patients, and possible links to ALK-TKI resistance were drawn [13]. EML4-ALK variant 1 was found to be the most frequent ALK variant, occurring in 43% of the investigated subjects, alongside EML4-ALK variant 3, which occurred in 40% of patients. ALK resistance mutations were much more common in variant 3, as compared to variant 1 (*p* = 0.023). In patients who received the third-generation ALK inhibitor lorlatinib, the EML4-ALK variant 3 was linked to a strikingly better PFS [14].

Increasing evidence suggests that NSCLC cells consequently develop resistance mechanisms against ALK-inhibitors in almost all cases, which makes it mandatory to follow up patients during the course of the disease by repeated molecular testing, especially in the case of tumor progression upon ALK-inhibitor treatment. 

In Figure 2, the complex manner of interaction of the EML4-ALK protein complex is illustrated, realized using a tandem affinity purification approach followed by mass spectrometry [11] (Figure 2).

To date, more than 6000 X-ray crystal structures have been discovered that are in the public domain of protein kinases [15]. An even larger number of three-dimensional proprietary structures are used by pharmaceutical companies for the discovery of new protein kinase inhibitors. Currently, about 175 protein kinase inhibitors that can be administered orally are being tested in clinical settings worldwide [16]. Close to 50 drugs that are directed against about 20 different protein kinases have already been approved by the Food and Drug Administration (FDA), having their points of action in about 20 different protein kinases [16,17]. Malignant cells are generally genomically unstable, and thus, resistance to protein kinase-targeting drugs occurs regularly over the disease course. As of today, it is not clear whether acquired resistance also occurs in protein kinase inhibitors when prescribed for inflammatory or autoimmune disorders [15]. All the different ALK fusion proteins feature a complex and multi-layered network of interaction with other proteins through a multitude of downstream pathways, like JAK/STAT, PI3K/AKT, or MEK/ERK [18,19]. When protein kinase inhibitors are administered over a longer time period, these complex models of interaction change in structure, leading to a dysregulation and, ultimately, acquired drug resistance [20]

## 2. Acquired ALK Resistance Mutations

Crizotinib, a first-generation ALK-TKI, was the first agent to be approved for clinical use. Crizotinib showed striking clinical efficacy when used as a therapeutic option in ALK-rearranged NSCLC. Recent follow-up data of clinical trials showed a response rate of >60% and a PFS of >12 months upon crizotinib therapy [21,22,23]. It has been clearly demonstrated for this agent that in nearly all patients showing good clinical response to treatment in the first place, resistance to the drug is acquired over time. Most often, secondary crizotinib resistance is due to acquired ALK gene mutations. Of note, de novo ALK resistance mutations, as well as pre-existing genetic aberrations leading to ALK-TKI therapy failure are generally rare (<3–5% of ALK-resistant NSCLC) [24]. Unlike epidermal growth factor (EGFR)-TKI resistance in NSCLC harboring EGFR mutations, where one single (EGFR T790M mutation) is outlined in about 60% of patients resistant to treatment, various ALK-resistance mutations (e.g., L1196M, I1171T/N/S, L1152P/R, F1174C/L/V, C1156Y/T, I1171T/N/S, S1206C/Y, G1269A/S, V1180L and 19 G1202R) are found in 20–25% of treatment-resistant subjects [25]. Second-generation ALK-TKIs, i.e., alectinib [26,27] and ceritinib [28], characterized by a different sensitivity-spectrum to ALK resistance mutations, have been approved as a treatment of NSCLC following resistance to crizotinib [25]. Various mechanisms of ALK-TKI resistance have been demonstrated so far, namely ALK gene amplification [29,30], activation of ALK via bypass signaling pathways [31,32], adopting different driver oncogenes like EGFR and BRAF [33], or insufficient drug penetration across the blood brain barrier and an enhanced expression of P-glycoprotein [34]. When these resistance mechanisms occur, other therapeutic options have to be implemented. Previous experience with crizotinib in clinical practice shows that nearly all ALK-positive patients are diagnosed with cerebral metastases sooner or later [35]. This is due to the lack of penetration of crizotinib to the blood brain barrier, even if patients still respond systemically to treatment. Moreover, only a small number of patients who develop brain metastases upon treatment with crizotinib develop ALK resistance mutations. Hence, the occurrence of brain metastases under crizotinib treatment strongly necessitated the development of second-generation ALK inhibitors. 

From a molecular standpoint, the more bulky and charged side chain of the ALK kinase is assumed to cause steric interference of most ALK inhibitors [29,36,37]. ALK F1174 mutations, for instance, are located very close to the C-terminus of the alpha C helix, most likely stabilizing and activating a conformation increasing the likelihood of ALK to bind to ATP [38,39].

In a study by Gainor et al., it was investigated how frequently ALK resistance mutations occur in a cohort of 51 patients with ALK-positive tumors, who had progressive disease upon treatment with crizotinib [30]. Tissue biopsies for this analysis were mostly acquired when patients still received crizotinib, or within one month after the stop of crizotinib therapy. In only 11 (20%) of biopsy samples, ALK resistance mutations were outlined. The most common ALK resistance mutations were L1196M and G1269A, but these were present only in 7% and 4% of the samples with crizotinib restistance, respectively [30]. Other mutations identified were C1156Y (2%), G1202R (2%), I1171T (2%), S1206Y (2%), and E1210K (2%). An interesting finding from the same study was that following treatment of second-generation ALK inhibitors, resistance mutations occurred more frequently [30]. Patients with disease progression upon treatment with ceritinib (*n* = 23), alectinib (*n* = 17), or brigatinib (*n* = 6) were investigated regarding ALK resistance mutations. Among 23 patients with ceritinib resistance, 21 (91%) had primarily received crizotinib. In nine patients, biopsies prior to initiation of ceritinib, or after crizotinib were also available, and only two of them showed on-target mechanisms of resistance. Overall, 54% of ceritinib-resistant tumor specimens harbored ALK resistance mutations, and 17% contained more than two different ALK resistance mutations, with G1202R (21%) and F1174C/L (16.7%) being the most common ones [30]. In the same study, the authors were also able to outline ALK C1156Y mutations in two (8%) of specimens, which lessens the response to ceritinib, as previously shown [36]. Moreover, a previously unknown ALK mutation, namely G1202del, was outlined in two tumor specimens (8%) [30]. Consecutively, in this study, Ba/F3 cells stably expressing EML4-ALK harboring the G1202del were engineered and treated with diverse ALK inhibitors. It was demonstrated that G1202del was associated with resistance to ceritinib, alectinib and brigatinib, while the efficacy of crizotinib was less affected by this deletion.

In addition, a cohort of 17 ALK-positive patients was analyzed, and all patients underwent repeated biopsies after they had developed progression upon treatment with alectinib. All 17 subjects had received crizotinib prior to alectinib [30]. In nine (53%) biopsy samples, ALK resistance mutations were outlined, the most common being G1202R, which occurred in 29% of cases. Interestingly, preclinical models have suggested alectinib to exert a considerable activity against L1196M, a known ALK-gatekeeper mutation [30], yet in this cohort of post-alectinib biopsies this mutation was observed in one subject as well. Another patient cohort of six ALK-positive subjects who had developed resistance to brigatinib was analyzed as well. ALK-resistance mutations were observed in five out of seven patients (71%). Similar to patients progressing upon ceritinib or alectinib treatment, the most common ALK resistance mutation was G1202R, which was outlined in three specimens [30]. Summing up their investigation, the authors pointed out that most patients received systemic chemotherapy (25%) or underwent enrolment into clinical trials (31%) after having developed resistance to ceritinib, alectinib and/or brigatinib. Of note, no consecutive therapy of any kind was administered in 38% of patients after they had progressed, or else no follow-up visit was done. According to a case report of a patient who relapsed upon brigatinib, the ALK-E1210K + D1203N double mutation was found [30]. However, it must be kept in mind that this patient had been treated with first-line crizotinib. Hence, no conclusions can be drawn between the findings in post-progression biopsies and the clinical response to sequential treatments. Across all three patient cohorts analyzed, 56% of ALK-positive patients progressed under treatment with second-generation ALK inhibitors (ceritinib: 54%; alectinib: 53%; brigatinib: 71%). Hence, ALK resistance mutations occurred at a significantly higher rate following therapy with the more potent second-generation ALK inhibitors when compared to a prevalence of ALK resistance mutations of only 20% in subjects who progressed under crizotinib [30].

Doebele and colleagues carried out a study aiming to outline resistance mechanisms to treatment with crizotinib [40]. Tissue from 14 ALK-positive NSCLC patients was obtained, all of whom progressed upon crizotinib therapy, as confirmed radiologically. Molecular analysis was performed with 11 tumor specimens. Four patients (36%) developed secondary ALK tyrosine kinase domain mutations. In two of these subjects, a new ALK mutation was outlined, encoding a G1269A amino acid substitution, which had previously been linked to crizotinib resistance in vitro. A newly occurring ALK copy number gain was seen in two patients, one of whom had an ALK resistance mutation [40]. One patient showed excessive growth of an EGFR mutant NSCLC, but an ALK gene rearrangement was not observed. Two patients featured mutation of KRAS, and one of these two patients did not have an ALK gene rearrangement. Interestingly, in one patient an ALK gene fusion negative tumor had newly developed, contrary to the baseline tumor sample. However, there was no identifiable alternative driver. Two patients retained ALK positivity in the absence of an identifiable resistance mutation [40]. Summing up these data about crizotinib resistance in ALK-positive NSCLC, crizotinib resistance is promoted by somatic mutations of the ALK kinase domain, by ALK gene fusion copy number gain and by newly emerging individual oncogenic mutations. 

Of note, according to a recent case report, a novel ROS1-FBXL17 (F-box and leucine-rich repeat protein 17) fusion, co-existing with CD74-ROS1 fusion was detected in a patient with lung adenocarcinoma, possibly improving sensitivity to crizotinib [41]. The authors point out that today only approximately 24 ROS1 fusion partners are known that exhibit crizotinib sensitivity. However, this non-reciprocal/reciprocal ROS1 translocation, containing a novel ROS1-FBXL17 fusion co-existing with the CD74-ROS1 fusion, was found in a Chinese patient suffering from lung adenocarcinoma, who responded well to treatment with crizotinib. Interestingly, PFS of this patient was 15.7 months, exceeding the highest PFS level reported among the Chinese population so far (14.2 months) [41]. The authors of this report suggest this particular ROS1-FBXL17 fusion to synergistically promote the sensitivity of the CD74-ROS1 fusion to crizotinib. This interesting case shows that crizotinib serves as a potential treatment option for patients with double ROS1 fusions, or non-reciprocal/reciprocal ROS1 translocation [41].

## 3. Epithelial-to-Mesenchymal Transition

The mechanism of epithelial-to-mesenchymal transition (EMT) has been recognized, with mounting evidence, as a tumorigenic driver, promoting the resistance to a variety of cytotoxic and targeted therapeutics, including EGFR-TKIs [25,33,42]. Previous research has found that patients with EGFR-TKI treatment who develop resistance caused by EMT have a much worse prognosis compared to subjects whose resistance is caused by T790M mutation [43]. Pathologic features of EMT include the loss of cell-to-cell contacts, and an enhancement in cell motility, resulting in a cell detachment from the parental epithelial tissue [44]. The switch to a rather mesenchymal-like phenotype makes cells more prone to migration, endorsing rapid tumor invasion and metastatic spread. Features of EMT were simultaneously found with ALK resistance mutations in a single tumor of a patient with ALK-rearranged lung cancer, who had developed resistance to ALK-TKI treatment [30]. It is as yet unclear, however, whether ALK-TKI resistant tumor cells that have adopted mesenchymal features develop ALK resistance mutations at the same time, or whether tumor cells with ALK resistance mutations and those undergoing EMT just co-exist in one and the same lesion [25]. Of note, there is no therapy for EMT-associated targeted drug resistance yet.

In a study by Fukuda et al., crizotinib-resistant tumor specimens that were obtained from one ALK-rearranged lung cancer patient were examined in depth [25]. The authors of this study found tumor lesions with a mesenchymal phenotype and ALK resistance mutations to co-exist in one and the same tumor lesion. Thus, EMT and ALK-rearrangement are evidently independent mechanisms, co-occurring in ALK inhibitor-resistant tumor specimens. As a next step, EML4-ALK lung cancer cell lines from humans were used to clarify in depth how exactly the induction of EMT contemporaneously with the acquisition of resistance to crizotinib takes place. Methods for overcoming ALK inhibitor resistance mediated by EMT were further illustrated in vitro and in vivo using laser capture microdissection (LCM) and digital PCR analysis.

The authors counted the copies of ALK L1196M, separately for epithelial- and mesenchymal-type tumor lesions [25]. To determine the presence of epithelial or mesenchymal features, tumor cells were immunohistochemically stained with E-cadherin and vimentin, respectively. E-cadherin-positive and vimentin-negative lesions, as well as E-cadherin-negative/vimentin-positive lesions, were specifically outlined. A copy number of > 12 copies of the ALK L1196M mutation could be observed in 1 µg of the whole DNA from tumor specimens with an epithelial phenotype. Conversely, the ALK L1196M mutation was hardly present in tumor cells that featured a predominantly mesenchymal pattern [25]. The EML4-ALK fusion gene is heterozygous, according to chromosomal analysis. In this study, digital PCR analysis revealed no increase in the number of copies of the ALK gene in tumors which harbored resistance to crizotinib. Only 10% of tumor cells featuring mesenchymal properties harbored the ALK L1196M mutation, which means that 90% of these mesenchymal-type tumor cells had developed crizotinib resistance, while no ALK mutation was present [25]. LCM was used to isolate also mesenchymal- and epithelial-type tumor lesions in metastatic samples. Notably, the ALK L1196M mutation was only observed in tumor lesions with epithelial features, being absent in the mesenchymal ones. Summing up the above-mentioned results, it becomes evident that EMT is an independent mechanism of resistance against ALK-targeted therapy [25].

As a next step, Fukuda et al. engineered a cell line derived from murine pleural effusions caused by the A925LPE3 lung cancer cell line [45]. The engineered cell line showed predominantly mesenchymal features. Treatment with crizotinib was performed in the novel (mesenchymal-like) cell line, as well as in the original A925LPE3 cell line. Crizotinib resistance was increased more than six-fold in the mesenchymal-like cell line as opposed to the parental A925LPE3 cells. Moreover, they were even cross-resistant to the next-generation ALK-TKIs alectinib, ceritinib and lorlatinib. Knockdown of ALK by means of a siRNA effectively reduced the viability of the parental A925LPE3 cells, but not of the engineered mesenchymal-type cells. Of note, any ALK resistance mutations were entirely absent in both the parental A925LPE3 and the novel cell line [25].

In a report from 2019, EMT was found to mediate also resistance to lorlatinib [46]. A 59-year-old male patient, who had been diagnosed with metastatic ALK-rearranged lung adenocarcinoma, received first-line treatment with crizotinib. He showed a partial response and PFS of 4.2 months. At the time when the disease progressed under crizotinib treatment, second line treatment with lorlatinib was initiated. Of note, at this point no repeated molecular analysis of tissue- or plasma-samples was performed. Sequential second-line treatment with lorlatinib at 75 mg daily was administered, and resulted in partial response, i.e., −78% according to RECIST criteria. After 6.9 months, further disease progression occurred, and a lung re-biopsy of the primary site was obtained [46]. Both the C1156Y and G1269A ALK mutations and the EML4-ALK variant 3 rearrangement (V3) were observed, both mutations being located at the same allele (i.e., compound mutation). Next, a cell line derived from this biopsy sample was engineered, and cell survival assays showed the cell line to be sensitive to lorlatinib. Thus, the C1156Y/G1269A compound mutation was most likely not the cause of lorlatinib resistance. Ba/F3 cells expressing either the EML4-ALK V3 with G1269A, C1156Y or the compound C1156Y/G1269A mutations were created in order to investigate in more detail the impact of this ALK compound mutation. Those Ba/F3 cells which expressed EML4-ALK together with the compound mutation, did not respond to lorlatinib therapy as well as the single mutations. The C1156Y/G1269A mutation also led to resistance to crizotinib, alectinib and entrectinib, but not brigatinib, according to in vitro tests [46]. Next, the aforementioned patient-derived cell line with good sensitivity to lorlatinib (MR57-S) was exposed to increasing dosages of lorlatinib until the tumor cells became resistant. The MR57 resistant (MR57-R) cell line was highly resistant to lorlatinib, and the presence of C1156Y and G1269A mutations was objectified in both cell lines [46]. Immunoblots of MR57-S and MR57-R cells showed ALK inhibition to result in the blocking of ERK, AKT and S6 phosphorylation, and induced apoptosis in the MR57-S cells. By sharp contrast, MR57-R cells still featured an abundance of extracellular signal-regulated kinases (ERK-), AKT serine/threonine kinases (AKT-) and ribosomal S6 kinase (S6) phosphorylation, as well as decreased apoptosis levels. Hence, an off-target mechanism of resistance via a bypass track is likely. The authors found that the morphology of MR57-S and MR57-R cells differed markedly. Consecutively, differential expression of EMT markers was assessed. High levels of E-cadherin, but no expression of N-cadherin and vimentin, were seen in MR57-S cells. In the MR57-R cells, however, E-cadherin expression was missing, but high levels of N-cadherin, Snail and vimentin were seen [46]. Clearly, the MR57-R cells had many properties that characterize the mesenchymal phenotype, and RNA sequencing of both cell lines served as an additional tool to confirm an aberrant expression of several genes related to EMT at the mRNA level: On the mRNA level, increased levels of vimentin, CDH-2 (N-cadherin), SNAIL, ZEB1, FGFR1 and TGFB1/2 expression were observed, alongside decreased levels of EPCAM, CDH-1 (E-cadherin), and ICAM1, as opposed to MR57-S cells. Additionally, phalloidin staining of actin microfilaments was performed in both cell lines. Lorlatinib-sensitive cells formed actin rings and showed proliferation in clusters, which is characteristic of an epithelial phenotype. MR57-R, on the other hand, contained actin stress fibers commonly observed in cells with a mesenchymal differentiation [46]. To confirm whether, at the time of tumor progression upon lorlatinib treatment, EMT-related properties occurred in the patient’s tumor, immunohistochemical analysis pre-crizotinib and at the time of progression upon lorlatinib was conducted. However, EMT features were not observed in the patient’s tumor at the time of progression under lorlatinib therapy. It is thus concluded that around the time of progression under lorlatinib, the EMT program is initiated, as shown in the cell culture experiments described above. Interestingly, another patient had become resistant to lorlatinib in the absence of any mutation causing ALK-TKI resistance. After treatment with crizotinib and ceritinib, this 58-year-old female, who was a lifetime non-smoker, received lorlatinib, and response lasted for 16 months. A cell line derived from a biopsy sample of this patient also featured mesenchymal features, and phalloidin staining confirmed the presence of actin stress fibers. When the patient’s biopsy samples pre-crizotinib and post-lorlatinib were immunohistochemically stained with EMT markers, an enhancement in vimentin expression in the post-lorlatinib sample was seen. This finding is suggestive of at least partial EMT in the tumor at the time when resistance to lorlatinib emerged [46].

Based on these findings, it is strongly assumed that EMT is an independent event rendering cells resistant to ALK-TKI treatment in the absence of ALK resistance mutations.

## 4. Genetic Assessment of ALK-Resistant Cancers

The most frequent mechanism by which tumors become resistant to second-generation ALK inhibitors is evidently the acquisition of ALK resistance mutations. Interestingly, in a study by Gainor et al. [30], 44% of biopsies from post-second-generation ALK-TKI treatment were found to be negative for any ALK mutations whatsoever. To investigate the possible role of other pathways leading to ALK-TKI resistance, tailored NGS was performed on post-ceritinib, post-alectinib and post-brigatinib biopsy samples in this analysis. Twenty-seven tumor specimens were included, and 15 (56%) of these exhibited aberrations in expression of one or more additional genes. Tumor protein P53 (TP53) mutations occurred most frequently, being present in 9 (33%) of biopsy samples. Notably, it was not possible to determine in retrospect when exactly these mutations occurred, hence, it is likely that they emerged before therapy with second-generation ALK-TKIs was initiated [30]. Missense mutations in DDR2 (L610F), BRAF (G15V), FGFR2 (F645L), MET (T992I), NRAS (A155T) and PIK3CA (G106V) could each be outlined in one (3.7%) of the tumor specimens, respectively, but none of these were co-occurring in one and the same biopsy sample. A MET T992I mutation was seen in one sample from this cohort (MGH040-2) that had previously been shown to occur at a low frequency in a variety of malignant cancers. Still, this gene variant was proven not to have the capacity of malignant transformation, and neither does it influence the phosphorylation pattern of MET [47]. MET T992I might not have acted as a relevant driver mutation in the above-mentioned study, either. However, one alectinib-resistant cancer did not feature resistance to ALK, yet, a PIK3CA G106V mutation was found, known to be linked to EGFR inhibitor treatment resistance in EGFR-mutant NSCLC [48]. Notably also in this patient, the time point at which the PIK3CA mutation emerged could not be exactly determined—hence, a relation to ALK-TKI resistance is debatable. According to previous data, a PIK3CA H1047R mutation could be outlined in a patient that had developed resistance to treatment with ceritinib [49,50], suggesting a possible connection. In the same study, an alectinib-resistant tumor specimen showed no ALK resistance mutation; however, a PIK3CA G106V mutation was found in this tumor. According to previous data, the PIK3CA G106V mutation is a gain-of-function mutation localizing to the p85/adaptor-binding domain of p110α, consecutively increasing AKT-phosphorylation [51]. PIK3CA mutations have been linked to acquired resistance to EGFR-TKIs in NSCLC featuring EGFR mutation [50].

Furthermore, Gainor and colleagues [30] sought to outline possible off-target resistance mechanisms to second-generation ALK inhibitors. For this purpose, they created six ceritinib-resistant cell lines derived from humans and performed NGS of 1000 pre-known cancer-related genes. One of these ceritinib-resistant cell lines which did not harbor ALK resistance mutations was treated with a MEK inhibitor. Of note, MEK-inhibitor therapy made cells sensitive to ceritinib therapy again, meaning that a de-novo activation of the MAPK signal transduction pathway possibly caused ceritinib restistance in this cancer specimen [40]. It has previously been reported that treatment with therapeutics targeting SRC, EGFR and PI3K re-sensitized cells to ALK inhibition, suggesting that these signaling pathways were the cause of ALK-TKI resistance in this model [49]. However, Gainor et al. did not outline any aberrations in the SRC-, EGFR- or PI3K pathway when performing the 1000-gene NGS analysis. Based on this observation, it can be concluded that ALK resistance mutations still pose the prime mechanism of resistance against second-generation ALK inhibitors. The off-target resistance mechanisms in the SRC kinase are often seen in ALK-positive and ALK-TKI-resistant NSCLC. ALK-resistant cells featured significantly higher levels of phosphorylation of SRC-related proteins, meaning that the SRC pathway as such may be a therapeutic target, with the aim of re-sensitizing patients to ALK inhibitor treatment [52]. The MEK kinase, which is located below ALK, could represent a resistance mechanism as well, because pre-clinical data suggest that a combination of MEK and ALK inhibition could possibly prevent, or at least delay, resistance [53]. Currently, clinical trials are investigating this issue (i.e., the NCT04292119 and NCT04055114 trials).

As an important note, the approach of liquid biopsy/cell-free DNA is becoming increasingly important in the assessment of ALK resistance, especially when secondary treatment failure to first-generation ALK-TKIs occurs. Contrary to tissue re-biopsies, liquid biopsy constitutes a safe and low-risk technique for disease monitoring, especially in patients with a poor performance status [54].

As a conclusion to this chapter, we want to highlight that targeted NGS for the identification of hot-spot mutations, single nucleotide variants and short indels, copy number variations, and gene fusions serves as a powerful tool to identify off-target resistance mechanisms, as well as secondary mutations in the ALK-kinase domain [30]. When additionally using morphological and IHC assessment of re-biopsy tissues, this allows for the identification of other resistance mechanisms, like EMT or the transformation of NSCLC to a small cell or undifferentiated phenotype.

## 5. Alectinib and Brigatinib

The second-generation ALK-TKI alectinib has its main points of action against ALK, as well as RET gene aberrations [55]. In comparison with crizotinib, alectinib has shown three times as much impact with respect to ALK inhibition in in vitro experiments, strongly enhancing apoptosis in AML4-ALK cell lines [40]. A variety of mutations linked to crizotinib resistance in the ALK tyrosine kinase domain (e.g., L1196M, G1269A, C1156Y, F1174L, 1151Tins, and L1152R) can be targeted effectively by alectinib [40,56]. Moreover, penetration of the blood–brain barrier by alectinib is good, as shown in an intracranial tumor model, where the growth of ALK-positive CNS lesions was effectively averted [57]. Several clinical trials have been carried out so far evaluating the effectiveness of alectinib. In a phase I/II study on 46 individuals diagnosed with ALK-rearranged untreated NSCLC, stable disease was reported for at least 6.5 months for seven subjects; two patients showed a complete response, and 41 patients responded partially to treatment [27]. Another study was carried out, where patients harboring crizotinib-resistant tumors were analyzed. In 55% of these individuals, an objective response was observed, comprising 2% complete response, 32% partial response and 20% not-confirmed partial response, while disease control was reported for 36% of patients. Of note, in the subgroup of patients with brain metastases (*n* = 21), an objective response was seen in 52% of these patients, whereas 29% had a complete response, 24% had a partial response and 38% of subjects experienced stable disease [26]. Reasonable activity of alectinib against brain metastases was reported in a pooled analysis of two phase II trials with crizotinib-resistant NSCLC patients [58,59,60,61]. The intracranial ORR reached 64%, while the systemic disease control rate was 90%, and the median duration of response amounted to 10.8 months. In the ALUR study, alectinib was proven superior as compared to platinum-based chemotherapy in late-stage NSCLC with resistance to crizotinib [62]. In this trial, the PFS was 7.1 months upon alctinib, when compared to chemotherapy (1.6 months). The ORR of CNS lesions amounted to 54.2%, and notably, chemotherapy warranted treatment discontinuation because of adverse events more often than alectinib (8.8% vs. 5.7%, respectively) [62]. Alectinib was also assessed as a first-line therapy, and compared to crizotinib in two phase III studies [63,64]. In the first patient cohort, the PFS upon crizotinib was 10.2 months, while PFS had not been reached at the time of data publication in the alectinib group. In the second trial, comparable results were observed, with a one-year event-free survival rate of 68.4% and 48.7% upon alectinib and crizotinib, respectively [63,64]. Looking at the available data on alectinib, superiority to crizotinib has evidently been proven, and in late-stage NSCLC, alectinib is also superior to platinum-based chemotherapy.

Brigatinib has shown broad-spectrum in vitro effectiveness against ALK, ROS1, insulin-like growth factor 1 receptor, EGFR and fms-like tyrosine kinase 3 [65]. According to a phase I/II trial, analyzing the safety and efficacy of brigatinib in a patient cohort with late-stage malignant disease, among them ALK-positive NSCLC, patients were sub-divided into five treatment groups [66]: First, ALK inhibitor-naïve patients with ALK-positive NSCLC;second, crizotinib-pre-treated ALK-positive NSCLC; third, EGFR T790M-positive NSCLC with EGFR-TKI resistance; fourth, patients suffering from other cancer entities; and fifth, NSCLC patients with CNS metastases, either crizotinib-naïve or post crizotinib. Only in NSCLC patients was response to brigatinib observed, with an ORR of 100% in group 1, 74% in group 2, 0% in group 3, 17% in group 4, and 83% in group 5 [66]. Of the crizotinib-pre-treated NSCLC patients, 72% featured an objective therapy response, while 100% of the crizotinib-naïve patients responded to brigatinib. Intracranial treatment response was, in total, 50%. In the ALTA phase II study, brigatinib was assessed in patients with crizotinib-refractory ALK-positive NSCLC [67]. The enrolled subjects were sub-divided into groups with either oral brigatinib at a fixed dosage of 90 mg daily, or 180 mg daily with a one-week dose escalation that also started at 90 mg. In these two patient groups, ORRs were 45% and 54%, respectively. A significant intracranial effect was achieved in 42% in group 1 and in 67% in group 2. The median PFS was 9.2 months and 12.9 months in groups 1 and 2, respectively, while the one-year OS was 71% and 80% [67]. Of note, the higher dose of brigatinib of 180 mg daily showed a consistently better effectiveness than 90 mg daily, while the safety profile remained reasonable. In the J-ALTA study, where a Japanese population with ALK-positive NSCLC was enrolled, the ORR and intracranial response was investigated as well [68]. The investigated patients had all progressed upon treatment with alectinib, with or without crizotinib. ORR was 30%, and intracranial response was modest with only 25%. In this trial, the median PFS was 7.3 months [68]. Patients with refractory secondary mutations in the ALK domain, like G1202R, I1171N and L1196M also responded well to brigatinib. The ALTA-1L study was conducted as an open-label phase III, international randomized trial, where the efficacy of brigatinib was compared to crizotinib in a large patient cohort (*n* = 275) with ALK-positive NSCLC, with all patients being naïve to ALK-TKI treatment [69]. Median PFS in the brigatinib group was significantly above the PFS in the crizotinib group (29.4 vs. 9.2 months; *p* < 0.001). ORR was 71% upon brigatinib and 60% upon crizotinib, while the intracranial response to treatment added up to 78% for brigatinib and 26% for crizotinib [69]. 

Both alectinib and brigatinib are obviously effective in treating ALK-positive NSCLC, especially after progression upon cizotinib. However, a limitation to both drugs, as well as to lorlatinib, is the fact that comparison has only been made with crizotinib, and not with later-generation ALK inhibitors.

## 6. Effectiveness of Lorlatinib

Lorlatinib is a third-generation, reversible, ATP-competitive and macrocyclic ALK- and ROS1-TKI, which can be orally administered [70]. As opposed to second-generation ALK inhibitors, central nervous system (CNS) penetration and overcoming pre-known secondary mechanisms of resistance in the ALK tyrosine kinase domain are properties unique to lorlatinib [71]. In preclinical experiments, lorlatinib was found to be more potent than TKIs directed against non-mutant ALK from previous generations, while retaining the capacity to act against many of the familiar mechanisms of resistance in the ALK tyrosine kinase domain that are acquired secondarily [38]. After having shown promising safety and efficacy in phase I and II clinical trials in treatment of patients with advanced-stage ALK- or ROS1-positive NSCLC [72], lorlatinib was approved for the treatment of pre-treated, advanced, ALK-positive NSCLC. 

Shaw and colleagues [71] carried out a study where the impact of lorlatinib specifically in relation to ALK resistance mutations in advanced NSCLC was analyzed. A total of 198 patients with ALK-positive NSCLC who had received at least one previous treatment with an ALK-TKI were included. Only 45 (24%) of the 189 patients harbored one or more ALK mutations in cfDNA. Plasma and tissue genotyping was consecutively applied to detect a large panel of different ALK mutations. In this study, the most frequent ALK mutations were G1202R/del (42%), L1196M (24%), F1174X (24%), G1269A (18%), and I1171X (11%) [71], for all of which sensitivity to lorlatinib treatment had previously been shown in preclinical models [30,37]. Among 59 patients who had received treatment with crizotinib prior to lorlatinib, the ORR was 73%, median duration of response was not reached, and median PFS was 11.1 months. Only 19% of patients in this subgroup harbored detectable ALK mutations. ORR did not quite differ between mutation-positive (73%) and mutation-negative (75%) patients [71]. With respect to the assessment of ALK mutation status, mutation-positive (73%) and mutation-negative (74%) subjects did not show any significant differences in ORR with lorlatinib either. Median PFS among patients who had received therapy with crizotinib before was not reached in the presence of ALK mutations, and was 12.5 months in the absence of ALK mutations. Duration of response was similar, irrespective of ALK mutation status, as well. Summing up these findings, lorlatinib is obviously highly effective in patients having received prior crizotinib as their only ALK-TKI, and effectiveness is independent of ALK mutation status [71]. In the 139 patients in this study who received lorlatinib after treatment with at least one ALK-TKI from the second generation, the ORR was 40%, the response lasted for a median time period of 7.1 months, and median PFS was 6.9 months. Twenty-six percent of patients in this cohort harbored identifiable ALK mutations, whereas 71% did not. Contrary to the findings in the post-crizotinib cohort, response rates after second-generation ALK-TKIs were dependent on ALK mutation status. The ORR was 62% among mutation-positive subjects, and 32% in the absence of ALK mutations. Hence, screening for ALK mutations in subjects who have previously been treated with one or more second-generation ALK-TKIs is feasible in order to outline patients that could particularly benefit from lorlatinib [71].

A global phase II study recently evaluated the effectiveness and safety of lorlatinib in patients with ALK-positive NSCLC [73]. Overall, 276 patients were included in this study, of whom 275 had received at least one dosage of lorlatinib. Thirty patients were treatment-naïve, 59 subjects had been administered crizotinib prior to lorlatinib, and 28 patients had been given ≥1 previous non-crizotinib ALK-TKI. Of note, eight (27%) of the 30 treatment-naïve subjects presented with CNS metastases at the time of enrolment in this study, while 133 (67%) in the cohort with prior therapy with one or more ALK-TKIs had brain metastases. Most of the investigated patients had an ECOG performance status of 0 or 1. In the treatment-naïve cohort, objective response could be observed in 90% of patients. For the individuals who harbored central nervous system metastases, objective intracranial response was observed in 66.7% of patients. Forty-seven percent of the patients in the subgroup with previous ALK-TKI treatment featured an objective response, and 63% of patients from this cohort with brain metastases had a measurable intracranial response [73]. Among the pre-treated patients, objective response was highest (69.5%) when having only received prior crizotinib, and it was 32.1% when having received one previous non-crizotinib ALK-TKI, and 38.7% in the cohort with previous treatment with two or more ALK-TKIs. According to this analysis, the most common adverse events of lorlatinib therapy were hypercholesterinaemia, which was diagnosed in 81% of the patients overall, followed by hypertriglyceridaemia (60%). The authors concluded that lorlatinib showed good overall and intracranial effectiveness in both treatment-naïve patients with ALK-positive NSCLC and in patients who progressed upon treatment with different ALK-TKIs [73]. Of note, this indicates that not only lorlatinib, but all second- and third-generation ALK inhibitors have been proven to be effective in penetrating the blood–brain barrier, markedly prolonging the time to progression when compared to crizotinib.

Yoda et al. conducted another study, where they specifically investigated sequential ALK inhibitor therapy and its relation to lorlatinib-resistant compound ALK mutations in lung cancer featuring ALK-positivity [74]. ALK compound mutations can appear in a cis or in trans configuration [75], and it has previously been described that under the pressure of lorlatinib treatment, these mutations are more often cis, which is linked to ALK-TKI treatment failure [76]. ALK and EGFR rearrangements may co-occur in one and the same tumor specimen [77], and it is already known that compound EGFR mutations feature a more aggressive behavior [67]. In what way exactly the mutual existence of ALK and EGFR compound mutations in NSCLC impacts tumor behavior still remains a matter of debate.

The authors sought to outline ALK mutations linked to resistance to lorlatinib treatment by implementing N-ethyl-N-nitrosourea (ENU) mutagenesis screening [78] of Ba/F3 models of ALK-positive cancer. Ba/F3 cell lines were used, and they either expressed wild type EML4-ALK as a model of ALK-TKI-naïve cancer, or else mutant EML4-ALK with one ALK resistance mutation, mimicking resistance following first- or second-generation ALK inhibitor treatment [74]. The Ba/F3 models feature differential sensitivity to various ALK inhibitors, with lorlatinib showing a striking impact against all models [30]. After the cells had been treated with ENU, artificially mutated cells were cultured with varying dosages of crizotinib or lorlatinib (100 nM–1000 nM), mimicking the administration of drugs in a clinical setting. Resistant clones were isolated, and DNA sequencing was performed to outline ALK kinase domain mutations. A variety of resistant clones after treatment with 300–600 nM of crizotinib emerged, harboring a variety of single ALK kinase domain point mutations, also including the majority of mutations causing resistance to crizotinib, as previously outlined in clinical settings [29,30,33,79]. By contrast, when treated with 300–600 nM of lorlatinib, the cells did not transform into resistant clones [74]. Notably, these drug concentrations are similar to the plasma levels in patients who received standard doses of lorlatinib. These findings are in line with previous studies, where a single ALK mutation responsible for therapy failure of lorlatinib could not be identified. Since many patients whose tumors progressed after administration of second-generation ALK-TKIs are consecutively treated with lorlatinib [30], the authors modeled this by once again performing ENU mutagenesis analysis of EML4-ALK expressing Ba/F3 cells, each featuring a familiar ALK resistance mutation after treatment failure of first- and second-generation ALK-TKIs (C1156Y, F1174C, L1196M, G1202R, and G1269A) [74]. After ENU mutagenesis, lorlatinib was added to the cells in culture in a concentration sufficiently preventing the outgrowth of one single mutant. A total of 12–49 clones that were resistant to lorlatinib were identified for each ALK mutant model. Mutagenesis of C1156Y or L1196M Ba/F3 cells developed clones resistant to lorlatinib that harbored a variety of ALK mutations. One of these mutations, ALK C1156Y/L1198F, was observed in a patient with resistance to lorlatinib previously [80]. Likewise, eight different compound ALK L1196M mutations emerged from the L1196M model, and two of these also comprised L1198 mutations [74]. The ALK G1202R/L1196M compound mutation was the only mutation that arose upon treatment with lorlatinib at the highest concentration (1000 nM), indicating that it represents a highly persistent lorlatinib resistance mutation, and was seen in a patient with acquired resistance to lorlatinib, as well. Notably, the emergence of the ALK G1202R/L1196M compound mutation upon lorlatinib treatment in this particular cell culture experiment cannot be generalized. We think that in clinical practice, after treatment failure of earlier-generation ALK inhibitors, lorlatinib is still the drug of choice, overcoming most, if not all, resistance mechanisms. Summing up these data, a vast variety of compound ALK mutations evidently mediate on-target resistance to lorlatinib [74]. As a next step, Yoda and colleagues sought to model lorlatinib resistance in vitro and in vivo. They treated sensitive H3122 cells with incremental dosages of lorlatinib over a time period of four months, until the cells became resistant. Three cell lines with resistance to lorlatinib (H3122 LR-A, LR-B and LR-C) were engineered and cultured in 1 µM lorlatinib. H3122 LR-A, LR-B and LR-C were all lorlatinib-resistant, as proven by cell viability assays [74], with none of them featuring a typical acquired mutation in the ALK tyrosine kinase domain. A mouse xenograft model of resistance to lorlatinib was engineered with cancer cells deriving from a sensitive EML4-ALK v1 cell line, namely MGH006. Mice carrying the tumors received therapy with lorlatinib, which led to a treatment response lasting for >50 days, as already reported by previous data [37]. With continuous therapy with lorlatinib, three out of six tumors became progressive, which was evidently a result of the development of resistance [74]. From the treatment-resistant tumors, three cell lines (MGH006 LR-B1, G3, and J2) were constructed. During culture, these three cell lines showed resistance to lorlatinib, and none of them carried an ALK resistance mutation. The conclusion from this experiment is that obviously no single ALK mutation confers resistance to lorlatinib.

In a study from 2020 by Shaw et al., first-line treatment with lorlatinib or crizotinib in advanced ALK-positive lung cancer was investigated [81]. In this global, randomized phase III trial, 296 treatment-naïve patients suffering from late-stage ALK-positive NSCLC were enrolled. Seventy-eight percent of patients in the lorlatinib-group were alive without disease progression at 12 months, whereas only 39% did not have progressive disease in the crizotinib group (*p* < 0.001). Objective responses were 76% in the lorlatinib-treated patients, and 58% in the crizotinib-treated group. In subjects who had been diagnosed with cerebral metastases, 82% and 23% showed an intracranial response, respectively. However, more grade 3 or 4 adverse events, primarily marked dyslipidemia, occurred upon treatment with lorlatinib than crizotinib (72% vs. 56%). In this interim analysis, conducted after approximately 75% of the expected number of progressive disease or decease had taken place, the superiority of lorlatinib regarding PFS and intracranial response was demonstrated [81].

However, an ORR comparable to that of lorlatinib (91.2%) was observed in the J-ALEX trial, comparing alectinib with crizotinib in ALK-positive NSCLC patients [63]. Ensartinib showed a clinical activity similar to lorlatinib, or that of second-generation ALK inhibitors, with an ORR of 80% and a PFS of 26.3 months in treatment-naïve NSCLC patients [82]. Likewise, the intracranial response of lorlatinib is obviously not superior to that of brigatinib, which is especially effective in the CNS because it contains a unique 5,5-dimethyl-1-pyrroline-N-oxide (DMPO) group, a feature that is not found in other ALK-TKIs, making it highly soluble in both water and fatty substances [83]. What makes lorlatinib special in clinical practice is the applicability especially in heavily pre-treated patients with intracranial progression [73]. A second feature unique to lorlatinib is its potency, which is higher than in other ALK-TKIs, at blocking the ALK tyrosine kinase. Thus, in crizotinib-pre-treated patients, lorlatinib is more effective in the presence of resistance mutations. Moreover, the ability of lorlatinib to overcome certain compound mutations, such as G1202R, makes this drug unique [84]. In another study, ALK mutations were outlined in circulating tumor cells from ALK-rearranged NSCLC patients who had progressive disease upon crizotinib or lorlatinib. The majority of mutations in different genes of ALK-independent pathways that obviously caused ALK-TKI resistance have been outlined in crizotinib-resistant patients. However, ALK compound mutations (ALKG1202R/F1174C and ALKG1202R/F1174L) have been found in one lorlatinib-resistant patient [85]. These data indicate that ALK compound mutations are often the cause of resistance to lorlatinib, leading to therapy failure of most later-generation ALK inhibitors.

As a conclusion, lorlatinib—like many of the post-first-generation ALK inhibitors—has almost exclusively been tested as a second line treatment, mostly following crizotinib. It is still a matter of debate as to whether lorlatinib should be implemented as a standard first-line medication, considering that sequential treatments, following earlier-generation TKIs, have produced the most reliable improvements in prognosis so far.

## 7. Conclusions

The subject of drug resistance mechanisms against first-, second- and third-generation ALK inhibitors is complex. Inherent ALK resistance mutations are only found in a proportion of patients with acquired resistance to ALK-TKI treatment. On-target resistance to the third-generation ALK inhibitor lorlatinib is primarily mediated by compound ALK mutations, according to the existing literature. A single ALK mutation leading to resistance against lorlatinib has not been outlined with certainty yet. For first- and second-generation ALK-TKIs, ALK mutations such as somatic kinase domain mutations still pose the prime mechanism of resistance to treatment. ALK gene fusion copy number gain, as well as different oncogenic driver mutations that emerge independently, constitute alternative pathways for cancer cells to develop ALK inhibitor resistance. Mesenchymal features are increasingly common in ALK-TKI-resistant tumor specimens, suggesting a role of EMT in resistance to ALK-TKIs as well.

## Figures and Tables

**Figure 1 cancers-13-00699-f001:**
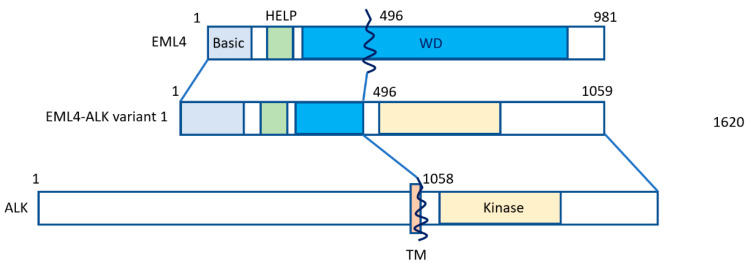
The EML4-ALK fusion gene. The N-terminal portion of EML4 is fused, containing the main region of EML4-ALK, i.e., the echinoderm microtubule-associated protein-like protein (HELP) domain, and part of the WD-repeat region to the intracellular region of ALK, which contains the tyrosine kinase domain. The transmembrane (TM) domain is not part of the final fusion product. Reproduced from Golding et al. [11].

**Figure 2 cancers-13-00699-f002:**
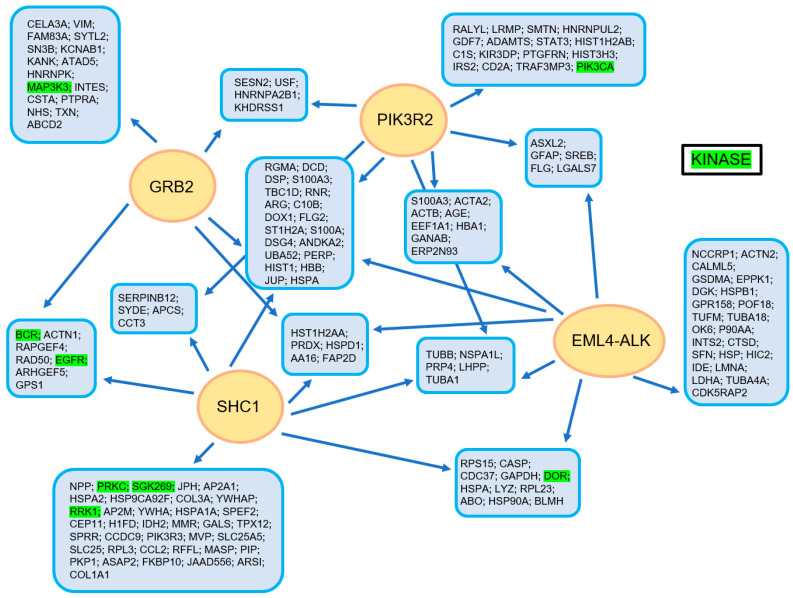
The EML4-ALK protein complex interaction model, as constructed using a tandem affinity purification approach with consecutive mass spectrometry. Reproduced from Golding et al. [11].

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
