# Peer review of "Current Knowledge about Mechanisms of Drug Resistance against ALK Inhibitors in Non-Small Cell Lung Cancer"

_cancers, 2021, doi:10.3390/cancers13040699_

Round 1

Reviewer 1 Report

I thank the authors and the editor for giving me the opportunity of reviewing this very interesting and proper review. It has a good structure and was easily to follow. Nevertheless, some minor issues should be clarified or corrected (don't worry about the length of the following - just arguing. Revision would not be very complicated/long):

  • The authors focus on molecular aberrations being responsible for resistance. Their review is complete in that sense, but one aspect needs to be clarified with a stronger tone: The experiences with crizotinib led to the knowledge that virtually all ALK-patients sooner or later get brain metastases, due to the lack of BBB penetration and even when systemic response is maintained. In contrast, and correctly stated by the authors, genetic aberrations responsible for resistance could only be detected in a minority of patients. The clinical need for second-generation inhibitors was definitely brain penetration, not (as with EGFR) efficacy against specific mutations. 
  • All mentioned second-gen inhibitors have tackled this problem, and all of them were highly superior to crizotinib regarding PFS (OS still immature) and intracranial responses. It is incorrectly stated in the manuscript that this was achieved by lorlatinib for the first time.
  • The introduction sounds to me a little bit too decent. The survival time for ALK NSCLC, which clinically always was considered one of the most aggressive molecular subtypes because of the inevitable brain involvement, has changed from an OS of a couple of months to a PFS of ≥30 months for alectinib, brigatinib, lorlatinib.
  • It sounds that lorlatinib is the next and consecutively necessary evolution of ALK inhibitors, which maybe is the case, but: There was much discussion after presentation of the lorlatinib data. Main critics - as with brigatinib - focused on the fact that all discussed inhibitors were only compared to crizotinib, and no direct comparison was made in first-line. Sequence approaches have (with fantastic data!) only be performed retrospectively. The "best-first" (if there were data) approach does not hold completely true when potential sequences are possible, especially when there might be molecular-guided therapies in progression. 

Superminor aspects:

  • Point 5: First, the word "effectiveness" is IMO not correct (I think "efficacy" is meant. Effectiveness would be efficacy in front of another variable, like time, costs...). Second, it is a little bit misleading that only lorlatinib efficacy data is presented. To get behind the recent discussions, the data should be shown in the light of/as compared with brigatinib and alectinib data, because all had the same comparator in the pivotal trials.
  • The affiliations are not linked correctly to the authors.

Author Response

Reviewer #1:

I thank the authors and the editor for giving me the opportunity of reviewing this very interesting and proper review. It has a good structure and was easily to follow. Nevertheless, some minor issues should be clarified or corrected (don't worry about the length of the following - just arguing. Revision would not be very complicated/long):

  • The authors focus on molecular aberrations being responsible for resistance. Their review is complete in that sense, but one aspect needs to be clarified with a stronger tone: The experiences with crizotinib led to the knowledge that virtually all ALK-patients sooner or later get brain metastases, due to the lack of BBB penetration and even when systemic response is maintained. In contrast, and correctly stated by the authors, genetic aberrations responsible for resistance could only be detected in a minority of patients. The clinical need for second-generation inhibitors was definitely brain penetration, not (as with EGFR) efficacy against specific mutations. 

RESPONSE: We thank Reviewer #1 for the careful revision of our manuscript. The first bullet point is indeed very important, explaining why the occurrence of brain metastases upon crizotinib treatment necessitated the development of second-generation ALK-TKIs. We have highlighted this in chapter 2, lines 136-142.

  • All mentioned second-gen inhibitors have tackled this problem, and all of them were highly superior to crizotinib regarding PFS (OS still immature) and intracranial responses. It is incorrectly stated in the manuscript that this was achieved by lorlatinib for the first time.

RESPONSE: Of course, Reviewer #1 is right about all second- and third-generation ALK-TKIs being superior to crizotinib, especially since they can penetrate the blood brain barrier. We have now included this statement in the manuscript, see lines 544-547.

  • The introduction sounds to me a little bit too decent. The survival time for ALK NSCLC, which clinically always was considered one of the most aggressive molecular subtypes because of the inevitable brain involvement, has changed from an OS of a couple of months to a PFS of ≥30 months for alectinib, brigatinib, lorlatinib.

RESPONSE: According to data from the literature, this remarkable improvement in PFS is true. However, to the best of our knowledge, no data on large patient cohorts, showing OS of ALK positive NSCLC without ALK TKI treatment, exists. We agree with Reviewer #1, that ALK positive NSCLC is one of the most aggressive NSCLC subtypes, and therefore, the improvement in PFS by ALK-TKIs has to be taken with a grain of salt. We have now addressed this issue in the introduction part, lines 61-66.

  • It sounds that lorlatinib is the next and consecutively necessary evolution of ALK inhibitors, which maybe is the case, but: There was much discussion after presentation of the lorlatinib data. Main critics - as with brigatinib - focused on the fact that all discussed inhibitors were only compared to crizotinib, and no direct comparison was made in first-line. Sequence approaches have (with fantastic data!) only be performed retrospectively. The "best-first" (if there were data) approach does not hold completely true when potential sequences are possible, especially when there might be molecular-guided therapies in progression. 

RESPONSE: We are grateful, that Reviewer #1 has pointed out this issue. At the end of chapter 6, we have now added a short paragraph where we address this possible limitation (lines 645-649).

Superminor aspects:

  • Point 5: First, the word "effectiveness" is IMO not correct (I think "efficacy" is meant. Effectiveness would be efficacy in front of another variable, like time, costs...). Second, it is a little bit misleading that only lorlatinib efficacy data is presented. To get behind the recent discussions, the data should be shown in the light of/as compared with brigatinib and alectinib data, because all had the same comparator in the pivotal trials.

RESPONSE: Regarding the headline of chapter 6 (formerly chapter 5), we disagree with Reviewer #1. We think that “effectiveness” is the correct term, since it means the ability of a drug or intervention (lorlatinib) to have a significant prognostic impact under real-life conditions. Efficacy, meanwhile, is rather used to describe an effect under ideal and controlled conditions. Since mainly clinical trials (and not in vitro experiments) are cited, we would like to keep the term “effectiveness”.

In order to also address brigatinib and alectinib, we have added a new chapter (chapter 5 – “Alectinib and Brigatinib”). See lines 408-481.

  • The affiliations are not linked correctly to the authors.

RESPONSE: Thank you for pointing out this mistake. The affiliations have now been adapted, so that they are correctly linked to each author.

Reviewer 2 Report

Thanks to Smolle E. et colleagues for having provided an interesting insight into “Current Knowledge about mechanism of Drug Resistance against ALK-inhibitors in NSCLC”. It represents a solid and well-written paper containing reference to relevant studies on this field. However, the article will be more accessible for the readers with some more tables summarizing many of the citated data.

The abstract
Line 16/17 – a good point. It is a difference between rearrangement and fusion. Not every rearrangement is a fusion.
Line 20 – In this sentence you mention that acquired mutations to be primary or secondary.
Do you mean as primary required and as secondary acquired?  
The sentence - as this - is misleading, since ALK-TKI resistance mutations can be primary = intristic, OR secondary = acquired. Intristic, inherent or de novo mutations describes the primary genomic profile before treatment. Acquired mutations are appearing after the diagnosis upon the treatment and on the course of the disease.

Keywords
– consider adding “acquired resistance mechanisms” and EMT.

The introduction chapter
In the introduction chapter, and before discussing the diverse drug resistance mechanism, it seems to be very essential to underline that that every single ALK-TKI has his own way to blockade ALK-kinase, and therefore provoke different forms of resistance. Different ALK-TKIs bind in the ATP‐binding pocket with varying contact sites resulting in unique profiles of inhibition of the ALK mutant variants observed in neuroblastoma patients as well as in ALK- TKI resistant NSCLC and inflammatory myofibroblastic tumor patients (Umapathy G. et al. Targeting anaplastic lymphoma kinase in neuroblastoma. APMIS, 2019, 127, 288–302. doi:10.1111/apm.12940). Furthermore, the profile of inhibition will be affected by different fusions variants of EML4-ALK and other different partners (Lin J.J. et al. Impact of EML4-ALK Variant on Resistance Mechanisms and Clinical Outcomes in ALK-Positive Lung Cancer. Journal of Clinical Oncology 36, 1199–1206. doi:10.1200/jco.2017.7 6.2294).

Line 44 – it is more crucial for the readers to underline that median OS for ALK-positive NSCLC is now about 7 years (Duruisseaux et al. Oncotarget 2017;8;21903-21917, Pacheco JM. et al. Natural History and Factors Associated with Overall Survival in Stage IV ALK-Rearranged Non–Small Cell Lung Cancer. J Thorac Oncol. 2019; 14(4): 691–700. doi: 10.1016/j.jtho.2018. 12.014), which is currently the best OS value for metastatic genomic defined NSCLC.

Both figures, 1. and 2., in the introduction chapter, are relevant, but the figure 2. lacks some follow up in the text. As the title is concerning drug resistance, more information should be addressed to drug resistance beyond acquired resistance in ALK-kinase domain and changing of phenotype, which the article describes extensively and sufficiently. In the modern era of genomic profiling some other mechanisms of acquired resistance should be mentioned in the wider context. The authors have mentioned though later (Lines 303-314) about the re-sensitizing to ALK-TKIs, while using therapeutics targeting SRC, EGFR and PI3K. There are just a couple interesting clues here to unfold. One of these is off-target resistance mechanism is Src-kinase, which is strategical kinase for many downstream of multiple resistance pathways in ALK-positive NSCLC. ALK resistant cells had higher phosphorylation levels of Src-related proteins suggesting that this pathway could be a therapeutic target in ALK resistant cell lines. Src inhibition (Saracatinib, Dasatinib) overcomes Alectinib resistance (Yoshida R. et al. Activation of Src signaling mediates acquired resistance to ALK inhibition in lung cancer. International Journal of Oncology 51, 1533–1540. doi:10.3892/ijo.2017.4140). Another point of reference is MEK-kinase localized below ALK in the signaling pathway, which can also represent resistance mechanism, similarly to that we have learned from BRAF-mutated tumors. There are preclinical data suggesting that combination of MEK and ALK inhibition may prevent or delay resistance (Hrustanovic G. et al. RAS-MAPK dependence underlies a rational polytherapy strategy in EML4-ALK–positive lung cancer. Nature Medicine, 2015: 21, 1038–1047. doi:10.1038/nm.3930), and clinicals studies addressed the issue (e.g., recruiting NCT04292119 and ongoing NCT04055114). Furthermore, multiple resistance mechanism can coexist, and it is not yet known whether a combination of ALK-TKI with chemotherapy (or immunotherapy) may work better in treating these patients. In the context of drug resistance, the findings from NCT03737994 study will be particularly important, informing about how data from preclinical studies finally work in humans.

Chapter 2
Line 61 – the chapter “ALK resistance mutations” may be rephased to “Acquired ALK resistance mutations” as it describes acquired mutations only. De novo ALK resistance mutations or preexisting genetic alterations appear to be rare: < 3 - 5 % of cases and in NSCLCs in general < 1% of cases, which is not the scope of the article (Lucena-Araujo A.R. et al. De novo ALK kinase domain mutations are uncommon in kinase inhibitor-naïve ALK rearranged lung cancers. Lung Cancer, 2016;99:17-22. doi:  0.1016/j.lungcan.2016.06.006).

Line 76 – Resistance mechanism in form of amplification of ALK-gene has only been observed under treatment with Crizotinib, i.e., first generation of ALK-TKI. Can you citate any data about ALK-amplification on second or third generation of ALK-TKIs?

Line 150/162 – please explain what is the reason to refer to action of Crizotinib in patients with ROS1 fusion?

Finally, in this chapter the issue of drug resistance to Brigatinib and Ensartinib should also be amended (as a current knowledge).

Chapter 3
Epithelial-to-mesenchymal transition is especially important resistance mechanism discussed by the authors, both in terms of both identifying this form of resistance (currently by IHC), and further treatment. There is quite a lot of preclinical and clinical data pointing out that EMT is independent ALK-TKI resistance mechanism, which may occur in absence of ALK- or other genes mutations or coexisting with them (Urbanska E.M. et al. Changing ALK-TKI-Resistance Mechanisms in Rebiopsies of ALK-Rearranged NSCLC: ALK- and BRAF-Mutations Followed by Epithelial-Mesenchymal Transition. International Journal of Molecular Sciences2020, 21, 2847. doi:10.3390/ ijms21082847).

Line 187 – please rephrase the sentence to clarify which criteria must be present for pathologist can diagnose EMT.

Chapter 4
Genetic assessment of ALK-resistant cancer is feasible as it was referred to the groundbreaking article of Gainor J.F. et al. (your reference 25). The decisive conclusion for readers will be to emphasize that extensive targeted next generation sequencing NGS using e.g., gene panel Oncomine Comprehensive Assay v3 (ThermoFisher Scientific; examines DNA / RNA for hot-spot mutations / SNVs + short indels, copy number variation / CNV and fusion in 161 cancer-relevant genes) for tumor material and gene panel Oncomine Lung Cell-Free Total Nucleic Acid Research Assay (ThermoFisher Scientific; examines hotspot mutations / SNVs + short divisions in 11 NSCLC-related genes such as ALK, BRAF, EGFR, ERBB2, KRAS, MAP2K1, MET, NRAS, PIK3CA, ROS1, and TP53, fusions in ALK, ROS1, and RET genes, MET exon 14 skipping, CNV for MET gene and detection limit / sensitivity of 0.1%) to ctDNA / RNA, will enable to identify resistance mechanism, both as secondary mutations in ALK-kinase domain, as well as off-target resistance. This should be supplemented by morphological and IHC assessment of the rebiopsy tissue to find other resistance mechanism, as EMT or another transformation to small cell or squamous phenotype.

In this chapter of “Genetic assessment of ALK-resistant Cancers”, using liquid biopsy (cfDNA) should be mentioned, although it is still not a standard, but probably the near future. However, there are data showing that this method can contribute to assess drug resistance (e.g., Madsen A.T. el al. Genomic Profiling of Circulating Tumor DNA Predicts Outcome and Demonstrates Tumor Evolution in ALK-Positive Non-Small Cell Lung Cancer Patients. Cancers, 2020, 12(4):947).

Chapter 5
Line 320 – The authors have rightly pointed out in the subsection “Effectiveness of Lorlatinib” that the drug´s CNS penetration and overcoming acquired resistance mutation is unique for Lorlatinib, citating findings from relevant clinical trials. However, it requires some more unfolding and description what it does Lorlatinib make to be such a unique drug. In treatment naïve patients in EXP1 cohort the ORR was 90%, which is impressive (Solomon B.J. et al. Lorlatinib in patients with ALK-positive non-small-cell lung cancer: results from a global phase 2 study. Lancet Oncol. 2018 Dec;19(12):1654-1667. doi: 10.1016/S1470-2045(18)30649-1). Nonetheless, the comparable ORR of 91,2 % was also observed in J-ALEX study in first- or second line setting (Hida T. et al. Alectinib versus Crizotinib in patients with ALK-positive non-small-cell lung cancer (J-ALEX): an open-label, randomized phase 3 trial. Lancet. 2017 Jul 1;390(10089):29-39. doi: 10.1016/S0140-6736(17)30565-2) in patients treated with Alectinib. Ensartinib has also shown to have comparable clinical activity to other second generation ALK TKIs (ORR 80 % and PFS 26,3 in treatment naïve patients) (Horn L. et al. Ensartinib (X-396) in ALK-Positive Non-Small Cell Lung Cancer: Results from a First-in-Human Phase I/II, Multicenter Study. Clin Cancer Res. 2018 Jun 15;24(12):2771-2779. doi: 10.1158/1078-0432.CCR-17-2398).
The same concerns the intracranial effect observed in treatment with second generation ALK-TKIs. E.g., Brigatinib is also highly effective in CNA because of unique DMPO group, a feature did not find in other ALK-TKis, which imparts a high-water solubility and lipophilicity, crucial for penetrance into the CNS system (Ando K. et al. Brigatinib and Alectinib for ALK Rearrangement-Positive Advanced Non-Small Cell Lung Cancer with Without Central Nervous System Metastasis: A Systematic Review and Network Meta-Analysis. Cancers, 2020 Apr 10;12(4):942. doi: 10.3390/cancers12040942)
The point rather is, especially important from the clinician´s perspective, that Lorlatinib can be effective in heavy pretreated patients, especially with progression in the brain, where it is hard to beat Lorlatinib. Solomon et al. (citated in the article, Solomon B.J. et al. Lorlatinib in patients with ALK-positive non-small-cell lung cancer: results from a global phase 2 study. Lancet Oncol. 2018 Dec;19(12):1654-1667. doi: 10.1016/S1470-2045(18)30649-1) showed in the EXP5-cohort that in patients treated with 3 prior ALK-TKIs the CNS response was 42%, which currently in such setting is the best outcome ever. Additionally, in all cohorts (EXP1, EXP2, EXP3A, EXP3B and EXP4) the intracranial response was higher than extracranial.
The second point, where Lorlatinib can be unique, is its innate potency higher than other first and second generation ALK-TKIs to blockade ALK-tyrosine kinase. It is clearly seen in the results presented by the authors (line 342/343), where Lorlatinib is effective in post Crizotinib-setting, indeed more effective in the presence of mutations. The same trend was observed in effectiveness after second-generation ALK-TKIs, where ORR was 32% in the absence of ALK- mutations (Line 352). It could mean on the one hand, that mechanisms of progression on Crizotinib are easy to overcome by such a potent ALK-TKI used after a relatively weak ALK-TKI, which Crizotinib seems to be. In that aspect the important issue is also the fact that the CNS relapse is the most frequent reason for treatment failure of Crizotinib, not constituting an obstacle to the Lorlatinib though. On the other hand, it can be interpreted that the greatest potency of Lorlatinib is seen as an ability to overcome some compound mutations, which probably explains the efficiency in heavy pretreated patients and highly suspected for developing more secondary mutations, especially G1202R, inclusive compound mutations. However, the complexity of potential pathways interfering in acquired resistance to Lorlatinib make that problem of resistance unfortunately continues with this third generation ALK-TKI (Redaelli S. et al. Lorlatinib Treatment Elicits Multiple On- and Off-Target Mechanisms of Resistance in ALK-Driven Cancer. Cancer Research, 2018: 78, 6866–6880).
Another aspect that describes high potency of Lorlatinib is the effect observed in EML4-ALK variant 3, perceived as worse responding to ALK-TKIs compared with other EML4-ALK variants (Christopoulos P. et al., Oncotarget, 2019; 10 (33):3093-3103, Tao H. et al. Distribution of EML4-ALK fusion variants and clinical outcomes in patients with resected non-small cell lung cancer. Lung Cancer. 2020 Nov;149:154-161).

Line 384 – you accurately refer to the paper written by Yoda S. et al. (Sequential ALK Inhibitors Can Select for Lorlatinib-Resistant Compound 550 ALK Mutations in ALK-Positive Lung Cancer. Cancer Discovery. 2018 Apr 12;8(6):714-29), where the authors though found one patient who progressed on Lorlatinib carrying a G1269A mutation, and one patient with a compound G1269A/G1202R mutation, indicating that it may represent a major challenge to Lorlatinib therapy. How do you interpret this finding? Also taking in consideration line 395/396.

Line 413 – addressing the issue of compound mutations: they can appear in-cis or in-trans configuration and it may be important, e.g., to treatment of EGFR compound mutations. Compound EGFR mutations display more aggressive behaviors. Do you have any comment on the importance of their mutual configuration in ALK-positive NSCLC to be overcome by Lorlatinib or other TKI, e.g., Repotrectinib?

Conclusions
Line 443 – do you mean “pre-known ALK resistance mutations” as preexisting? You may consider rephrasing the sentence using intristic (or inherent, de novo).

Thank you.

Author Response

Reviewer #2:

Thanks to Smolle E. et colleagues for having provided an interesting insight into “Current Knowledge about mechanism of Drug Resistance against ALK-inhibitors in NSCLC”. It represents a solid and well-written paper containing reference to relevant studies on this field. However, the article will be more accessible for the readers with some more tables summarizing many of the citated data.

The abstract

Line 16/17 – a good point. It is a difference between rearrangement and fusion. Not every rearrangement is a fusion. Line 20 – In this sentence you mention that acquired mutations to be primary or secondary.
Do you mean as primary required and as secondary acquired?  
The sentence - as this - is misleading, since ALK-TKI resistance mutations can be primary = intristic, OR secondary = acquired. Intristic, inherent or de novo mutations describes the primary genomic profile before treatment. Acquired mutations are appearing after the diagnosis upon the treatment and on the course of the disease.

RESPONSE: We thank Reviewer #2 for carefully going through our manuscript an for all the precious inputs. In lines 32-33 in the abstract, we have now deleted the term “acquired”, and only wrote “primary or secondary” instead, because Reviewer #2 is right, of course, that “acquired” is synonymous with “secondary”.

Keywords
– consider adding “acquired resistance mechanisms” and EMT.

RESPONSE: We consider this a good suggestion, and have accordingly added “acquired resistance mechanisms”, and “epithelial-mesenchymal transition” to the keywords (lines 38-39).

The introduction chapter

In the introduction chapter, and before discussing the diverse drug resistance mechanism, it seems to be very essential to underline that that every single ALK-TKI has his own way to blockade ALK-kinase, and therefore provoke different forms of resistance. Different ALK-TKIs bind in the ATP‐binding pocket with varying contact sites resulting in unique profiles of inhibition of the ALK mutant variants observed in neuroblastoma patients as well as in ALK- TKI resistant NSCLC and inflammatory myofibroblastic tumor patients (Umapathy G. et al. Targeting anaplastic lymphoma kinase in neuroblastoma. APMIS, 2019, 127, 288–302. doi:10.1111/apm.12940). Furthermore, the profile of inhibition will be affected by different fusions variants of EML4-ALK and other different partners (Lin J.J. et al. Impact of EML4-ALK Variant on Resistance Mechanisms and Clinical Outcomes in ALK-Positive Lung Cancer. Journal of Clinical Oncology 36, 1199–1206. doi:10.1200/jco.2017.7 6.2294).

Line 44 – it is more crucial for the readers to underline that median OS for ALK-positive NSCLC is now about 7 years (Duruisseaux et al. Oncotarget 2017;8;21903-21917, Pacheco JM. et al. Natural History and Factors Associated with Overall Survival in Stage IV ALK-Rearranged Non–Small Cell Lung Cancer. J Thorac Oncol. 2019; 14(4): 691–700. doi: 10.1016/j.jtho.2018. 12.014), which is currently the best OS value for metastatic genomic defined NSCLC.

RESPONSE: We think that this is a very good point. Therefore, we have now added the above-mentioned statements to the introduction chapter: lines 74-86 and lines 60-63.

Both figures, 1. and 2., in the introduction chapter, are relevant, but the figure 2. lacks some follow up in the text. As the title is concerning drug resistance, more information should be addressed to drug resistance beyond acquired resistance in ALK-kinase domain and changing of phenotype, which the article describes extensively and sufficiently. In the modern era of genomic profiling some other mechanisms of acquired resistance should be mentioned in the wider context.

RESPONSE: It would indeed be a good addition to the introduction chapter, if some general information about the molecular biology and process of drug resistance against ALK-TKIs be added. Hence, we have briefly addressed this issue in the last paragraph of the introduction part (lines 98-113). Unfortunately, since the manuscript is already quite long, we cannot go into further detail with the various mechanisms of ALK-kinase domain resistance.

The authors have mentioned though later (Lines 303-314) about the re-sensitizing to ALK-TKIs, while using therapeutics targeting SRC, EGFR and PI3K. There are just a couple interesting clues here to unfold. One of these is off-target resistance mechanism is Src-kinase, which is strategical kinase for many downstream of multiple resistance pathways in ALK-positive NSCLC. ALK resistant cells had higher phosphorylation levels of Src-related proteins suggesting that this pathway could be a therapeutic target in ALK resistant cell lines. Src inhibition (Saracatinib, Dasatinib) overcomes Alectinib resistance (Yoshida R. et al. Activation of Src signaling mediates acquired resistance to ALK inhibition in lung cancer. International Journal of Oncology 51, 1533–1540. doi:10.3892/ijo.2017.4140). Another point of reference is MEK-kinase localized below ALK in the signaling pathway, which can also represent resistance mechanism, similarly to that we have learned from BRAF-mutated tumors. There are preclinical data suggesting that combination of MEK and ALK inhibition may prevent or delay resistance (Hrustanovic G. et al. RAS-MAPK dependence underlies a rational polytherapy strategy in EML4-ALK–positive lung cancer. Nature Medicine, 2015: 21, 1038–1047. doi:10.1038/nm.3930), and clinicals studies addressed the issue (e.g., recruiting NCT04292119 and ongoing NCT04055114). Furthermore, multiple resistance mechanism can coexist, and it is not yet known whether a combination of ALK-TKI with chemotherapy (or immunotherapy) may work better in treating these patients. In the context of drug resistance, the findings from NCT03737994 study will be particularly important, informing about how data from preclinical studies finally work in humans.

RESPONSE: We are grateful for these remarks of Reviewer #2. The above-mentioned literature is now cited in a new section at the end of chapter 4 (lines 388-407).

Chapter 2

Line 61 – the chapter “ALK resistance mutations” may be rephased to “Acquired ALK resistance mutations” as it describes acquired mutations only. 

RESPONSE: According to this suggestion, we have rephrased the headline of chapter 2.

De novo ALK resistance mutations or preexisting genetic alterations appear to be rare: < 3 - 5 % of cases and in NSCLCs in general < 1% of cases, which is not the scope of the article (Lucena-Araujo A.R. et al. De novo ALK kinase domain mutations are uncommon in kinase inhibitor-naïve ALK rearranged lung cancers. Lung Cancer, 2016;99:17-22. doi:  0.1016/j.lungcan.2016.06.006).

RESPONSE: This circumstance is now stated in the first paragraph of chapter 2 (lines 121-123).

Line 76 – Resistance mechanism in form of amplification of ALK-gene has only been observed under treatment with Crizotinib, i.e., first generation of ALK-TKI. Can you citate any data about ALK-amplification on second or third generation of ALK-TKIs?

RESPONSE: This is a good point. However, after a careful literature research, to the best of our knowledge, ALK resistance because of ALK gene amplification has only been described in the context of crizotinib treatment so far.

Line 150/162 – please explain what is the reason to refer to action of Crizotinib in patients with ROS1 fusion?

RESPONSE: We thought that after all the pre-existing data, clearly demonstrating that crizotinib treatment over a longer time period leads to drug resistance in the majority of patients, it would be reasonable to mention the efficacy of crizotinib also against ROS1. Since also for first-line treatment, other ALK-TKIs than crizotinib may be used, in case of an additional ROS1 fusion in an ALK-positive patient, crizotinib as a first-line treatment should be considered.

Finally, in this chapter the issue of drug resistance to Brigatinib and Ensartinib should also be amended (as a current knowledge).

RESPONSE: Not much is known about resistance mechanisms against brigatinib yet. We have already addressed the G1202R mutation, which was outlined in brigatinib-resistant subjects (lines 182-184). Additionally, we have added one additional report about a brigatinib-resistant patient lines 188-191. To the best of our knowledge, ensartinib was only investigated as a tool to overcome resistance to other ALK-TKIs (mainly crizotinib), but no in-depth analysis about resistance mechanisms in patients with therapy failure to ensartinib has yet been published.

Chapter 3

Epithelial-to-mesenchymal transition is especially important resistance mechanism discussed by the authors, both in terms of both identifying this form of resistance (currently by IHC), and further treatment. There is quite a lot of preclinical and clinical data pointing out that EMT is independent ALK-TKI resistance mechanism, which may occur in absence of ALK- or other genes mutations or coexisting with them (Urbanska E.M. et al. Changing ALK-TKI-Resistance Mechanisms in Rebiopsies of ALK-Rearranged NSCLC: ALK- and BRAF-Mutations Followed by Epithelial-Mesenchymal Transition. International Journal of Molecular Sciences2020, 21, 2847. doi:10.3390/ ijms21082847).

Line 187 – please rephrase the sentence to clarify which criteria must be present for pathologist can diagnose EMT.

RESPONSE: Indeed, the pathologic features of EMT should be explained. Hence, we have now described, what histologic properties are characteristic for EMT (lines 234-237).

Chapter 4

Genetic assessment of ALK-resistant cancer is feasible as it was referred to the groundbreaking article of Gainor J.F. et al. (your reference 25). The decisive conclusion for readers will be to emphasize that extensive targeted next generation sequencing NGS using e.g., gene panel Oncomine Comprehensive Assay v3 (ThermoFisher Scientific; examines DNA / RNA for hot-spot mutations / SNVs + short indels, copy number variation / CNV and fusion in 161 cancer-relevant genes) for tumor material and gene panel Oncomine Lung Cell-Free Total Nucleic Acid Research Assay (ThermoFisher Scientific; examines hotspot mutations / SNVs + short divisions in 11 NSCLC-related genes such as ALK, BRAF, EGFR, ERBB2, KRAS, MAP2K1, MET, NRAS, PIK3CA, ROS1, and TP53, fusions in ALK, ROS1, and RET genes, MET exon 14 skipping, CNV for MET gene and detection limit / sensitivity of 0.1%) to ctDNA / RNA, will enable to identify resistance mechanism, both as secondary mutations in ALK-kinase domain, as well as off-target resistance. This should be supplemented by morphological and IHC assessment of the rebiopsy tissue to find other resistance mechanism, as EMT or another transformation to small cell or squamous phenotype.

RESPONSE: In lines 401-407, we have now highlighted this important conclusion of chapter 4, based on the work of Gainor and colleagues.

In this chapter of “Genetic assessment of ALK-resistant Cancers”, using liquid biopsy (cfDNA) should be mentioned, although it is still not a standard, but probably the near future. However, there are data showing that this method can contribute to assess drug resistance (e.g., Madsen A.T. el al. Genomic Profiling of Circulating Tumor DNA Predicts Outcome and Demonstrates Tumor Evolution in ALK-Positive Non-Small Cell Lung Cancer Patients. Cancers, 2020, 12(4):947).

RESPONSE: This issue has now been addressed in the article, citing the suggested work by Madsen et al. (lines 396-400).

Chapter 5

Line 320 – The authors have rightly pointed out in the subsection “Effectiveness of Lorlatinib” that the drug´s CNS penetration and overcoming acquired resistance mutation is unique for Lorlatinib, citating findings from relevant clinical trials. However, it requires some more unfolding and description what it does Lorlatinib make to be such a unique drug. In treatment naïve patients in EXP1 cohort the ORR was 90%, which is impressive (Solomon B.J. et al. Lorlatinib in patients with ALK-positive non-small-cell lung cancer: results from a global phase 2 study. Lancet Oncol. 2018 Dec;19(12):1654-1667. doi: 10.1016/S1470-2045(18)30649-1). Nonetheless, the comparable ORR of 91,2 % was also observed in J-ALEX study in first- or second line setting (Hida T. et al. Alectinib versus Crizotinib in patients with ALK-positive non-small-cell lung cancer (J-ALEX): an open-label, randomized phase 3 trial. Lancet. 2017 Jul 1;390(10089):29-39. doi: 10.1016/S0140-6736(17)30565-2) in patients treated with Alectinib. Ensartinib has also shown to have comparable clinical activity to other second generation ALK TKIs (ORR 80 % and PFS 26,3 in treatment naïve patients) (Horn L. et al. Ensartinib (X-396) in ALK-Positive Non-Small Cell Lung Cancer: Results from a First-in-Human Phase I/II, Multicenter Study. Clin Cancer Res. 2018 Jun 15;24(12):2771-2779. doi: 10.1158/1078-0432.CCR-17-2398).
The same concerns the intracranial effect observed in treatment with second generation ALK-TKIs. E.g., Brigatinib is also highly effective in CNA because of unique DMPO group, a feature did not find in other ALK-TKis, which imparts a high-water solubility and lipophilicity, crucial for penetrance into the CNS system (Ando K. et al. Brigatinib and Alectinib for ALK Rearrangement-Positive Advanced Non-Small Cell Lung Cancer with Without Central Nervous System Metastasis: A Systematic Review and Network Meta-Analysis. Cancers, 2020 Apr 10;12(4):942. doi: 10.3390/cancers12040942)
The point rather is, especially important from the clinician´s perspective, that Lorlatinib can be effective in heavy pretreated patients, especially with progression in the brain, where it is hard to beat Lorlatinib. Solomon et al. (citated in the article, Solomon B.J. et al. Lorlatinib in patients with ALK-positive non-small-cell lung cancer: results from a global phase 2 study. Lancet Oncol. 2018 Dec;19(12):1654-1667. doi: 10.1016/S1470-2045(18)30649-1) showed in the EXP5-cohort that in patients treated with 3 prior ALK-TKIs the CNS response was 42%, which currently in such setting is the best outcome ever. Additionally, in all cohorts (EXP1, EXP2, EXP3A, EXP3B and EXP4) the intracranial response was higher than extracranial.
The second point, where Lorlatinib can be unique, is its innate potency higher than other first and second generation ALK-TKIs to blockade ALK-tyrosine kinase. It is clearly seen in the results presented by the authors (line 342/343), where Lorlatinib is effective in post Crizotinib-setting, indeed more effective in the presence of mutations. The same trend was observed in effectiveness after second-generation ALK-TKIs, where ORR was 32% in the absence of ALK- mutations (Line 352). It could mean on the one hand, that mechanisms of progression on Crizotinib are easy to overcome by such a potent ALK-TKI used after a relatively weak ALK-TKI, which Crizotinib seems to be. In that aspect the important issue is also the fact that the CNS relapse is the most frequent reason for treatment failure of Crizotinib, not constituting an obstacle to the Lorlatinib though. On the other hand, it can be interpreted that the greatest potency of Lorlatinib is seen as an ability to overcome some compound mutations, which probably explains the efficiency in heavy pretreated patients and highly suspected for developing more secondary mutations, especially G1202R, inclusive compound mutations. However, the complexity of potential pathways interfering in acquired resistance to Lorlatinib make that problem of resistance unfortunately continues with this third generation ALK-TKI (Redaelli S. et al. Lorlatinib Treatment Elicits Multiple On- and Off-Target Mechanisms of Resistance in ALK-Driven Cancer. Cancer Research, 2018: 78, 6866–6880).
Another aspect that describes high potency of Lorlatinib is the effect observed in EML4-ALK variant 3, perceived as worse responding to ALK-TKIs compared with other EML4-ALK variants (Christopoulos P. et al., Oncotarget, 2019; 10 (33):3093-3103, Tao H. et al. Distribution of EML4-ALK fusion variants and clinical outcomes in patients with resected non-small cell lung cancer. Lung Cancer. 2020 Nov;149:154-161).

RESPONSE: In lines 624-644, we have cited the above-mentioned articles, explaining the exact features making lorlatinib unique. We are very thankful to Reviewer #2 for providing this input, alongside the respective literature links, for us.

Line 384 – you accurately refer to the paper written by Yoda S. et al. (Sequential ALK Inhibitors Can Select for Lorlatinib-Resistant Compound 550 ALK Mutations in ALK-Positive Lung Cancer. Cancer Discovery. 2018 Apr 12;8(6):714-29), where the authors though found one patient who progressed on Lorlatinib carrying a G1269A mutation, and one patient with a compound G1269A/G1202R mutation, indicating that it may represent a major challenge to Lorlatinib therapy. How do you interpret this finding? Also taking in consideration line 395/396.

RESPONSE: In line 591-595, we have written two sentences as an interpretation of the results of the study by Yoda et al.

Line 413 – addressing the issue of compound mutations: they can appear in-cis or in-trans configuration and it may be important, e.g., to treatment of EGFR compound mutations. Compound EGFR mutations display more aggressive behaviors. Do you have any comment on the importance of their mutual configuration in ALK-positive NSCLC to be overcome by Lorlatinib or other TKI, e.g., Repotrectinib?

RESPONSE: We thank Reviewer #2 for pointing this out. In lines 550-557 we have briefly commented on this topic.

Conclusions

Line 443 – do you mean “pre-known ALK resistance mutations” as preexisting? You may consider rephrasing the sentence using intristic (or inherent, de novo).
Thank you.

RESPONSE: Accordingly, we have changed the term “pre-known” to “inherent”.

Reviewer 3 Report

Smolle and colleagues have made a well written review about TKIs and resistance in ALK positive lung cancer. They show that different TKIs are available in the current treatment and that some treatment options are still under investigation. Resistance to TKIs is present and many different mechanisms have been described.

However I do have comments on this review:

  1. The story line is not clear to me. What is the message you want to bring? What do you want to add on top on the current reviews already present? Make this more clear by adapting this review and us other headings and subheadings. I suggest to subdivide to different TKIs or to different mechanisms or resistance. Now they are both mentioned but not in a good storyline.
  2. In the introduction you mention different methods to detect ALK fusions (line 35-37). I miss however newer techniques such as RNA based testing such as Archer, Nanostring and whether it is important which test should be used.
  3. Figure 1 is not clear to me; The meaning of WD is not mentioned. Where do the numbers count for? I think you should add the exons and what the differences are in the breakpoint, influencing whether it is a variant 1 or 3 and so on.
  4. Figure 2 does not add anything to the paper. There is no explanation of this figure so, I do not understand where I am looking at
  5. I miss items about resistance mechanisms caused by the fusion partner and the importance of the different EML4-ALK variants.
  6. In chapter 2 treatment with different TKI, resistance on cell lines and resistance on biopsies are mentioned randomly. Make better subheadings to give a better insight on what you are talking about.
  7. The subheading of chapter 2 is resistance mechanisms. However the first item you describe is the use of TKIs. Make this more clear
  8. Chapter 3 about EMT is mostly about EMT in crizotinib resistance. I suggest to put this item under resistance mechanisms in crizotinib.
  9. I miss a good description about the effect of ceritinib, alectinib and brigatinib and what should be given as standard treatment now.
  10. In part 5 about lorlatinib you describe a lot about resistance in cell lines. However there are many case reports about compound mutations in patients. I suggest to add that as well.

Author Response

Reviewer #3:

Smolle and colleagues have made a well written review about TKIs and resistance in ALK positive lung cancer. They show that different TKIs are available in the current treatment and that some treatment options are still under investigation. Resistance to TKIs is present and many different mechanisms have been described.

However I do have comments on this review:

  1. The story line is not clear to me. What is the message you want to bring? What do you want to add on top on the current reviews already present? Make this more clear by adapting this review and us other headings and subheadings. I suggest to subdivide to different TKIs or to different mechanisms or resistance. Now they are both mentioned but not in a good storyline.

RESPONSE: We thank Reviewer #3 for carefully revising our article, and for the valuable feedback. Based on the suggestions of the other two reviewers, we have already re-structured and re-phrased major parts of the manuscript, e.g. we have added a new chapter on alectinib and brigatinib (chapter 5). All the new text passages / paragraphs are marked in red. We hope, that Reviewer #3 might agree now, that the story line has improved.

  1. In the introduction you mention different methods to detect ALK fusions (line 35-37). I miss however newer techniques such as RNA based testing such as Archer, Nanostring and whether it is important which test should be used.

RESPONSE: This is indeed a good point made by Reviewer #3. Accordingly, we have now mentioned the Archer®FusionPlex® panel as a valid alternative to other techniques for the detection of ALK rearrangements (lines 51-55).

  1. Figure 1 is not clear to me; The meaning of WD is not mentioned. Where do the numbers count for? I think you should add the exons and what the differences are in the breakpoint, influencing whether it is a variant 1 or 3 and so on.

RESPONSE: “WD” cannot be written out; it is a domain name as such. In the figure legend, we explain “the WD-repeat region to the intracellular region of ALK (…)”. The numbers indicate the base pairs, i.e. the location of the gene on the chromosome. Although we agree with Reviewer #3, that a more detailed description of the molecular properties would be a valid addition to figure 1, we cannot go into more detail with the molecular biology, since our article is already very long. Moreover, this manuscript is focusing on clinical studies, rather than on molecular details.

  1. Figure 2 does not add anything to the paper. There is no explanation of this figure so, I do not understand where I am looking at

RESPONSE: We agree with Reviewer #3, that figure 2, at first glance, is a little confusing. With this image, we wanted to depict the complexity of the ALK-EML4 protein, and its countless interaction points with other proteins / kinases. We hope that – although our article is mainly oriented clinically – we may keep figure 2 in the manuscript as it is.

  1. I miss items about resistance mechanisms caused by the fusion partner and the importance of the different EML4-ALK variants.

RESPONSE: We thank Reviewer #3 for pointing this out. Hence, we have added some information about this issue to the introduction chapter (lines 78-86).

  1. In chapter 2 treatment with different TKI, resistance on cell lines and resistance on biopsies are mentioned randomly. Make better subheadings to give a better insight on what you are talking about.

RESPONSE: Although we are thankful for this input, we disagree with Reviewer #3 in this bulletpoint. Since nearly all studies we review in chapter 2 are about ALK-TKI resistance upon treatment with crizotinib, we do not think it is reasonable to further divide this chapter with subheadings.

  1. The subheading of chapter 2 is resistance mechanisms. However the first item you describe is the use of TKIs. Make this more clear

RESPONSE: As mentioned above, the majority of studies about ALK-TKI resistance have investigated resistance upon crizotinib treatment (and the potential impact of consecutively administered second- and third-generation ALK inhibitors). Therefore, we thought it would be necessary to provide a brief introduction about crizotinib at the very beginning of chapter 2.

  1. Chapter 3 about EMT is mostly about EMT in crizotinib resistance. I suggest to put this item under resistance mechanisms in crizotinib.

RESPONSE: Most of the literature on ALK-TKI resistance deals with resistance in the context of crizotinib. However, the additional impact of EMT, in our opinion, warrants a separate chapter. Since Reviewer #3 also suggested, that we put subheadings in chapter 2, we consider it unreasonable to fuse chapters 2 and 3 altogether.

  1. I miss a good description about the effect of ceritinib, alectinib and brigatinib and what should be given as standard treatment now.

RESPONSE: We think, this is a good point. Hence, we have added a new chapter (chapter 5), going into more detail with alectinib and brigatinib (lines 408-481).

  1. In part 5 about lorlatinib you describe a lot about resistance in cell lines. However, there are many case reports about compound mutations in patients. I suggest to add that as well.

RESPONSE: Data about ALK compound mutations in patients do exist, however, to the best of our knowledge, not in the context of lorlatinib. We have cited one study, where data from a patient with an ALK compound mutation who developed resistance to lorlatinib, is described (Recondo et al.). Moreover, we have added a new paragraph to the chapter (lines 637-644), where we have cited one other case report about ALK compound mutations and lorlatinib resistance (Pailler et al.).

Round 2

Reviewer 3 Report

The manuscript improved a lot. Thank you for your comments and it is reasonable to disagree on some points with me. Good that you had this discussion about relevance of figures.

I have no further comments